# Traditional Thai Massage Promoted Immunity in the Elderly via Attenuation of Senescent CD4+ T Cell Subsets: A Randomized Crossover Study

**DOI:** 10.3390/ijerph18063210

**Published:** 2021-03-19

**Authors:** Kanda Sornkayasit, Amonrat Jumnainsong, Wisitsak Phoksawat, Wichai Eungpinichpong, Chanvit Leelayuwat

**Affiliations:** 1Biomedical Sciences Program, Graduate School, Khon Kaen University, Khon Kaen 40002, Thailand; kandas@kkumail.com; 2The Centre for Research and Development of Medical Diagnostic Laboratories (CMDL), Faculty of Associated Medical Sciences, Khon Kaen University, Khon Kaen 40002, Thailand; amonrat@kku.ac.th (A.J.); wisiph@kku.ac.th (W.P.); 3School of Medical Technology, Faculty of Associated Medical Sciences, Khon Kaen University, Khon Kaen 40002, Thailand; 4Department of Microbiology, Faculty of Medicine and Graduate School, Khon Kaen University, Khon Kaen 40002, Thailand; 5School of Physical Therapy, Faculty of Associated Medical Sciences, Khon Kaen University, Khon Kaen 40002, Thailand

**Keywords:** traditional Thai massage (TTM), the elderly, CD4+ T subsets, the natural killer group 2, member D (NKG2D), interleukin-17 (IL-17)

## Abstract

The beneficial physiological effects of traditional Thai massage (TTM) have been previously documented. However, its effect on immune status, particularly in the elderly, has not been explored. This study aimed to investigate the effects of multiple rounds of TTM on senescent CD4+ T cell subsets in the elderly. The study recruited 12 volunteers (61–75 years), with senescent CD4+ T cell subsets, who received six weekly 1-h TTM sessions or rest, using a randomized controlled crossover study with a 30-day washout period. Flow cytometry analysis of surface markers and intracellular cytokine staining was performed. TTM could attenuate the senescent CD4+ T cell subsets, especially in CD4+28^null^ NKG2D+ T cells (*n* = 12; *p* < 0.001). The participants were allocated into two groups (low < 2.75% or high ≥ 2.75%) depending on the number of CD4+28^null^ NKG2D+ T cells. After receiving TTM over 6 sessions, the cell population of the high group had significantly decreased (*p* < 0.001), but the low group had no significant changes. In conclusion, multiple rounds of TTM may promote immunity through the attenuation of aberrant CD4+ T subsets. TTM may be provided as a complementary therapy to improve the immune system in elderly populations.

## 1. Introduction

Traditional Thai massage (TTM) is a type of manual therapy in complementary and alternative medicine that applies pressure to the skin and soft tissues of the body along the ten meridian lines [1]. The beneficial effects of TTM include a reduction in stress-related parameters, improvement in physical relaxation, recovery from myofascial pain, and promotion of well-being [1,2,3,4,5,6]. Previous studies investigated the associations between massage and molecular mechanisms, such as immunomodulatory effects, inflammatory pathways, neuronal and pain modulation. There are many inflammatory responses after the application of minimal pressure on muscles, including the migration of several immune cells to the affected area associated with a complex cell signaling system [7]. Complex interactions between skeletal muscle and the immune system regulate muscle regeneration [8] and stimulate effector T cells due to mechano-transduction [7,8,9,10]. Regarding functional immune cells, these T cells could be mechanosensing and activate immune responses [11]. In spite of this, there are few studies on the mechanisms of massage-related immunity improvement. Nowadays, application of TTM for health improvement is recommended as a complementary and alternative therapy along with standard treatment in diabetes mellitus (DM) patients [12] and rehabilitation from sports injuries [13]. However, it has not been well-studied for health maintenance and immunological improvement in the elderly.

The age-associated decline in immunity (referred to as immunosenescence) is characterized by systemic low-grade inflammation due to increases in aberrant immune cell functions, impairment of immune cell developments and elevated pro-inflammatory cytokine productions, potentially leading to several chronic inflammatory diseases [14]. A well-known example is the activation of T helper (CD3+CD4+) lymphocytes, which are members of T cell subpopulations in adaptive immune responses. Normally, this cell type is activated via the signaling of T cell receptors (TCR) co-operating with a co-stimulatory molecule, CD28 [15]. In many conditions, such as chronic inflammations and infectious or autoimmune diseases [16,17,18,19,20], CD28 expressions were reduced or diminished (CD4+CD28^null^); however, surrogate co-stimulatory molecules were present on these cells, resulting in alterations in immune responses [14,16,18,21,22,23,24]. Previous studies in the elderly suggested that the natural killer group 2 member D (NKG2D) played a role as one of the surrogate co-stimulatory molecules of CD4+CD28^null^ T cells, leading to higher pro-inflammatory cytokine production [16,18,19,20,23,24,25,26]. This CD4+ T cell subset may be considered as a pathological T cell subpopulation and a biomarker of immunosenescence.

In this study, we hypothesized that TTM may provide beneficial effects on immune status by reducing the senescent CD4+ T subsets and may also inhibit pro-inflammatory cytokine production, especially that of IL-17 and IFN-γ, in the elderly. We designed a randomized controlled study with a 30-day wash out crossover period. Flow cytometric analysis was performed to analyze the immunophenotype and cytokine productions of CD4+ T cells and their subsets by using multi-colored surface and intracellular staining.

## 2. Materials and Methods

### 2.1. Participants

The proposal of this study, with a crossover design, was approved by the Khon Kaen University Ethical Committee Review Board (HE 602018) in April 2017 in accordance with the 1964 Helsinki Declaration. The research protocol of this study was registered at the Thai Clinical Trials Registry (TCTR 20190326002), under the title “Effect of Traditional Thai Massage on Immune Status in Aging (TMIA)”. All protocols were performed according to relevant guidelines and regulations. Twelve elderly participants aged 60–80 years with a history of elevated senescent CD4+ T cells were recruited from a sub-district health-promoting hospital between December 2017 and April 2019. They gave informed consent prior to participating in the study. None of the participants had severe diabetic complications, infectious diseases, autoimmune diseases, hematological diseases, malignancies, or inflammatory illness. Candidates with contraindications to traditional Thai massage, according to the previous report [1], were excluded. All the participants fully complied with the study protocol and remained until the end of the study. A statistical program, G*power 3.1 (Heine University, Dusseldorf, Germany), was used to determine sample size, where the α and power (1 − β) were set as 0.05 and 0.8, respectively [27].

### 2.2. Interventions

Traditional Thai massage (TTM) was applied in a total of six weekly 1-h sessions throughout the whole body (T), with a crossover resting period (R). Each session was performed at the same time of day once a week. Twelve participants underwent a TTM period and a rest period for 37 days, with a 30-day washout crossover intervention, as mentioned. A qualified massage therapist, certified in massage therapy by the Ministry of Public Health, carried out the massage on all participants individually. Two participants were treated simultaneously; while one received a massage session, the other one was assigned to rest on another massage bed. A standard TTM protocol was applied throughout the whole body. Moderate to deep pressure of the thumb and palm along ten meridian basal lines of the whole body was applied. The massage was conducted on lower limbs, back, neck, head, and upper limbs following the order of supine position, lying on the left side, and lying on the right side, as described in previous studies [1]. Each participant was followed up with an interview and measurement of their blood pressure 1 month after the last intervention.

### 2.3. Study Design

A randomized and crossover design was chosen because self-paired effects could reduce variability due to interindividual differences [28]. The lottery method was employed by the staff in this study. Both procedures—TTM, and resting—were numbered on separate slips of paper of the same size, shape, and color, folded and shuffled in a container. Blind sampling was administered. The washout period of 30 days was estimated according to a previous study of the effectiveness of acupuncture on lymphocyte proliferation in the elderly [29]. All participants were randomly allocated into different starting sessions: 6 for TTM and 6 resting. A crossover trial of 6 weekly sessions of 1-h TTM and rest for 37 days with a 30-day washout period was performed. Fresh blood samples were collected by a licensed medical technologist before intervention (pre) and within 1 day after the last round of the 6 sessions (post). The study design is illustrated using the CONSORT 2010 flowchart diagram in Figure 1.

Twelve Thai elderly (60–80 years) were purposively sampled. The crossover trial encompassed 6 weekly sessions of 1-h TTM and resting for 37 days with a 30-day washout period. Blood was collected from each participant 4 times including before TTM (pre-TTM), before resting (pre-resting) and after receiving 6 sessions of TTM (post-TTM) and after receiving 6 resting sessions (post-resting). All participants were analyzed by flow cytometric analysis.

### 2.4. Blood Collection and Chemical Parameters

At the time of blood collection, 10 mL peripheral blood samples were drawn and allocated with or without different anticoagulants, including 1 mL in ethylene diamine tetra-acetic acid (EDTA) for complete blood count (CBC) and the glycated hemoglobin A1c (HbA1c), 7 mL in heparin for surface and intracellular staining and 2 mL in a clotted-blood tube for blood chemistry testing (creatinine, alkaline phosphatase (ALT), and lipid profiles; total cholesterol (TC), triglyceride (TG), low-density lipoprotein cholesterol (LDL-C) and high-density lipoprotein cholesterol (HDL-C)). All blood parameters were provided by the Medical Technology Laboratory, Faculty of Associated Medical Sciences, Khon Kaen University.

### 2.5. Measurement of Cell Surface Markers of CD4+ T Cells and Their Intracellular Cytokine Production

One hundred microliters of fresh peripheral blood from the heparinized tube (within 24 h of collection) were used for staining of cell surface markers using monoclonal antibodies (mAbs) conjugated with different fluorochromes as follows: CD3-FITC (Immunotool, Friesoythe, Germany), CD28-PE-Cy7 (BD Pharmingen, San Diego, CA, USA), CD4-PE (Immunotool, Friesoythe, Germany), NKG2D-APC, clone 149,810 (R&D systems, Minneapolis, MN, USA), isotype control PE-Cy7 (BioLegend, San Diego, CA, USA) and isotype control APC (R&D systems, Minneapolis, MN, USA). After incubation at room temperature for 15 min in the dark, red blood cells were lysed using the FACS lysing solution (BD Bioscience, San Jose, CA, USA) for 15 min and then washed with 1X PBS. Then, the samples were analyzed by the flow cytometer. Gating populations of CD28 and NKG2D were based on the isotype controls of PE and APC, respectively. Representative immunophenotyping analyses were shown as percentages of the scatter plot (one point as one cell). Positive quartiles of CD3+ and CD4+ were calculated as percentages of CD4+ T cells among total lymphocytes. Then, CD4+ T cells were gated and divided into 2 groups depending on CD28 expression: namely, CD3+4+28+ (CD4+28+ T cell) and CD3+4+28^null^ (CD4+28^null^ T cell). Finally, the percentages of NKG2D-expressing cells in those T subsets were determined on CD4+ (CD4+NKG2D+), CD4+28+ (CD4+28+NKG2D+), and CD4+28^null^ (CD4+28^null^NKG2D+). Gating populations of CD28 and NKG2D were based on the isotype controls. An example of gating of these CD4+ T cell subpopulations and analysis was shown in Appendix A.

For a functional assay performed by stimulating T cells, 7 mL of heparinized blood was activated by 100 ng/mL PMA (Sigma Aldrich^®^, Saint Louis, MO, USA) and 1000 ng/mL ionomycin calcium salt (Sigma Aldrich^®^, St. Louis, MO, USA), followed by staining with multi-color mAbs against surface markers including CD3-PE (ImmunoTools, Friesoythe, Germany), CD4-APC-Cy7 (BioLegend, San Diego, CA, USA), CD28-PE-Cy7 (BD Pharmingen^TM^, San Diego, CA, USA), NKG2D (CD314)-APC (R&D systems, Minneapolis, MN, USA), isotype control-PE-Cy7 (BioLegend, San Diego, CA, USA) and isotype control-APC (R&D systems, USA) prior to the intracellular staining of IL-17-PerCP-Cy5.5 (BD Pharmingen^TM^, San Diego, CA, USA) and IFN-γ-FITC (BioLegend, San Diego, CA, USA). Gating the percentages of IFN-γ and IL-17 on CD4+28^null^ NKG2D+ T cells was based on medium (un-stimulated) and isotype control. Representative percentages of both IL-17 and IFN-γ intracellular producing CD4+28^null^ NKG2D+ T cells are shown in Appendix A.

All procedures are described elsewhere in previous studies [20,26]. The flow cytometry service was provided by the Research Instrument Center (RIC), Khon Kaen University, Thailand.

### 2.6. Statistical Analysis

Flow cytometry data were analyzed by using the BD FACSDivas^TM^ software (BD Biosciences). All the data in Table 1 are expressed as mean ± standard deviation (SD). The graphs were made using GraphPad Prism software (GraphPad Software Inc., La Jolla, CA, USA). Normal distribution was tested by the Shapiro–Wilk test. All analyses were executed under an intention-to-treat approach. A multivariate test was used to determine overall significant differences between groups. Then, a two-way repeated-measures ANOVA was used to determine significant main effects and interactions in outcome variables (the percentages of cell populations) at pre- and post-TTM and the rest period (two periods, two treatments, 2 × 2). These tests are based on linearly independent pairwise comparisons among the estimated marginal means. The Bonferroni post hoc test was used for multiple comparisons. Changes in concentrations of intracellular cytokines within the TTM phase were compared pre- and post-test using the Wilcoxon Signed Rank test. The level of statistical significance was set at *p*-value < 0.05.

## 3. Results

### 3.1. Participant Characteristics

The participants were 12 elderly Thai (including 9 females and 3 males) aged between 61 and 75 years (67.7 ± 5.1). Their data were recorded from December 2017 to April 2019. The baseline characteristics of the participants with health history records indicated that some of them had one or more underlying conditions such as hypertension (17%), diabetes mellitus (DM) (42%), and dyslipidemia (25%). Moreover, they had elevated senescent CD4+ T cell subsets, including CD4+28^null^, CD4+NKG2D+, CD4+28+NKG2D+, and CD4+28^null^NKG2D+ (Table 1).

### 3.2. Alteration of CD4+ T Cell Subsets after TTM

The percentage of CD4+ T cell subsets is represented by mean ± SD compared between pre- and post-intervention for both the TTM and resting groups. These cells, including CD4+28+, CD4+28^null^, CD4+NKG2D+, and CD4+28^null^NKG2D+, showed significant differences (*p* = 0.035, 0.039, 0.044, and 0.006, respectively) (Table 2). Additionally, post hoc analyses revealed that the statistical power for this study was 0.17 for CD4+ T cells, whereas the power exceeded 0.99, 0.98, 0.96, 0.77, and 0.99 for CD4+28+, CD4+28^null^, CD4+NKG2D+, CD4+28+NKG2D+, and CD4+28^null^NKG2D+, respectively. The findings indicated a moderate to large effect size (range 0.33–17.6). Thus, there was adequate power (>0.80) at the moderate to large effect size level. The significant *p* value between pre- and post-TTM, effect size and post hoc power calculations are displayed in Table 3.

The responses to TTM of CD4+ T cell subsets were revealed in comparisons within the groups of TTM. The percentages of total CD4+ T cells were not significantly different between TTM and resting. However, the percentages of conventional CD4+ T cells (CD4+28+ T) increased significantly after post-intervention with repeated massage (*p* = 0.016; Table 3 and Figure 2). Conversely, senescent CD4+ + T cells, including CD4+28^null^, CD4+NKG2D+, CD4+28+NKG2D+, and CD4+28^null^NKG2D+, were decreased significantly in post-intervention after massage once per week with a total of 6 sessions (*p* = 0.023, 0.001, 0.012, and <0.001, respectively; Table 3 and Figure 2). Although there was a considerable difference between baselines (pre-TTM and rest sessions), there were no significant differences between them for all parameters. Similarly, the pre- and post-resting comparison revealed no significant difference in all parameters (Figure 2).

### 3.3. The Reduction in High Percentages of CD4+CD28^null^NKG2D+ T Cells after Multiple Rounds of TTM

To confirm the effectiveness of multiple rounds of TTM on reducing senescent CD4+ T cells, especially CD4+CD28^null^NKG2D+, the participants were divided into two groups—those with high and low percentages of CD4+C28^null^NKG2D+ T cells based on the median of the percentages of these cells before massage (pre-TTM). The median of pre-TTM was 2.75% which was used as a cut-off. The average percentage of these cells in the high group was 8.3 ± 3.1% (*n* = 6, range 3.2–12.6%), and in the low group was 1.4 ± 0.7% (*n* = 6, range 0.5–2.3%). The high group had highly significant decreases in % CD4+28^null^ NKG2D+ T cells after receiving TTM with *p* < 0.001 (Table 3 and Figure 3). The effectiveness of multiple interventions of the high group compared with other groups and times was significantly different (*p* < 0.001; Table 2). Moreover, post hoc analysis achieved high power for type II error probability (1 − β) with the value of 1.00 at the large effect size level (17.6) of the high group (*n* = 6; Table 3).

### 3.4. Effect of Multiple TTM on Intracellular Cytokines IFN-γ and IL-17 of the High Group for CD4+28^null^ NKG2D+ T Cells

The effect of multiple rounds of TTM on the percentages of IFN-γ and IL-17 producing T cells in the high group (*n* = 6) showed no significant differences (*p* = 0.344 and 0.075, respectively). However, decreasing mean levels were evident after repeated massages. Representative data were shown by median ± SE (Figure 4). The levels of IFN-γ pre and post were 6.3 ± 1.4 and 2.7 ± 0.7, respectively. The production of IL-17 pre and post was 5.0 ± 1.9 and 3.1 ± 0.9, respectively.

## 4. Discussion

Traditional Thai massage (TTM), a common manual therapy in complementary and alternative medicine, is a technique that applies pressure to the skin and soft tissues on ten meridian lines called “Sen Sib”. Stimulation of blood flow in the vessels has been described as energy flowing throughout the body along these lines [1]. However, there are few studies that show the effect of TTM in immunity promotion. This study preliminarily reported the effectiveness of multiple rounds of traditional Thai massage (TTM) on reducing senescent CD4+ T cells related to their functions in producing low-grade inflammation. Our crossover study was purposefully designed to control for individual immunological variations. In order to demonstrate the effect of TTM, 12 elderly participants who had senescent CD4+ T cell subpopulations representative of immunosenescence were recruited for study.

From the flow cytometry data, the percentages of CD4+ T cell were not different when compared between pre- and post-TTM. However, a reduction in the percentage of CD4+ T subsets was found in participants after receiving TTM, including CD4+28^null^, CD4+NKG2D and especially, CD4+28^null^NKG2D+, suggesting that TTM may be involved in the improvement of immunity by decreasing aberrant CD4+ T subsets. In aging and immune disorders, increased aberrant CD4+CD28^null^ occurs [14,16,18,22,23,24], with predominantly NKG2D expression [19,20,25,26]. Several studies have indicated that these senescent CD4+ T cells are associated with pathogenic features in chronic inflammatory diseases, including autoimmune diseases [18,19,22], cardiovascular diseases (CVD) [18,20], diabetes mellitus (DM) [26], and advanced age [16,18,23,24,25]. These immunosenescence cells can also produce proinflammatory cytokines, such as IFN-γ, IL-6, TNF-α, and IL-17 [14,18,19,20,26]. Their presence may be linked to faster progression of age-related diseases. Therefore, accumulation of CD4+28^null^ T cells has been proposed as a biomarker and prognostic factor in chronic immune-mediated diseases and the immunosenescence of aging [18,24]. Similarly, the expansion of NKG2D expression on CD4 T cells has been the target of immunotherapy, which is currently under investigation [19]. The therapeutic effect of blocking NKG2D to decrease IL-17 in arthritis (an autoimmune disease) has been demonstrated [30]. Notably, promoting reduction of these immunosenescence cells could reduce systemic chronic inflammation in the elderly.

A previous study in mice using massage-like stroking demonstrated improvement in immunity by increasing thymocytes including CD4+CD8+, CD4+CD8-, and CD4-CD8+ cells in the lymphoid organs (thymus and spleen) [31]. Additionally, the effect of electro-acupuncture stimulation on antifebrile points (ST36) could affect T cell compositions by increasing lymphocytes in the blood, particularly CD4+ T cells [32]. Study on the efficacy of Swedish massage on the number of CD4+, CD8+, CD25+, and CD56+ cells in healthy young participants reported a higher number of these cells after massage once weekly for 5 weeks [33]. However, the effect of light TTM on CD4+ and CD8+ T cells had no change in colorectal cancer patients [34]. Outcomes on efficiency for immunity improvement may be different depending on study population, times of intervention, techniques and pressure applied.

Additionally, the effect of multiple rounds of TTM on subjects with high percentages of CD4+28^null^ NKG2D+ T cells was more evident than those in the low group. Activating receptor NKG2D of CD4+ T cells showed anomalous appearances, with rising expressions in chronic inflammation and autoimmunity [19,20,25,26]. The elderly with CD4+28^null^NKG2D+ at more than 2.75% had significant decreases in their level of these cells after receiving multiple rounds of TTM. Conversely, those with CD4+28^null^NKG2D+ lower than 2.75% had no change. Therefore, we suggest that repeated TTM might be useful to apply in elderly populations who have a particularly high level of CD4+28^null^NKG2D+ cells. Previously, CD4+28^null^ NKG2D+ T cells producing both IFN-γ and IL-17 cytokines were functionally related to elderly populations with multiple risk factors of CVD [20] and T2DM [26]. As mentioned, these cells could be associated with the severity of diseases. Among complementary and alternative medicines (CAMs), Chinese acupuncture, Thai massage, stretching exercises, and herbalism were used for healthcare management in diabetes mellitus [12].

Although the effect of TTM in modulating immune cells and functions has been poorly studied, the mechanisms by which massage therapy activates potentially beneficial immunomodulatory pathways have continuously been explored [7,35]. A possible mechanism of massage in general is that it alters the signaling pathways involved in the inflammatory process, reducing secondary injury, nerve sensitization and collateral sprouting, which results in recovery from damage and a reduction in pain through mechano-transduction [7]. Moreover, the mechanism of the mechanical stimuli from the extracellular matrix (ECM) is through the stimulation of integrin receptors’ functions via focal adhesion complexes (FACs) to the muscle cytoskeleton. Sequentially, activation of focal adhesion kinase (FAK) phosphorylation initiates progrowth signaling pathways within the muscle fiber [10]. For immune cells, they are recruited to play the critical role of adaptation for maintenance via response to massage stimuli in tissue homeostasis processes. Mechanosensation also modulates immune cell phenotypes and modifies cellular physiology, which eventually determines the functionality of the cells [11]. From our study, immunomodulation is potentially associated with mechano-transduction via the proinflammatory cytokine IFN-γ and IL-17 of CD4+ T subset, especially in the subset with high NKG2D expression. Furthermore, the therapeutic effect of multiple-round TTM may occur due to tissue homeostasis.

There are limitations in this study. Firstly, we did not monitor the participants’ daily physical activities which may have interfered with our experiments. We were only able to encourage them to continue their typical daily activities without making any intentional abrupt changes to their routines. Secondly, since purposive sampling includes participants from a specific population, the results may not be suitable when applying them to general populations. However, these results can still provide valuable information and offer useful recommendations for elderly people with underlying conditions. We acknowledge that the effect of TTM may only be clearly demonstrated in subjects who have high percentages of aged immunosenescence cells. Hence, this led to our study design. Finally, a larger sample size should be recruited for further functional study. The number of participants for investigation of IFN-γ and IL-17 should be 14 and 18, respectively, to reach statistical significance at the 0.05 level and 80% power using a randomized, controlled crossover study.

## 5. Conclusions

This study is the first report to demonstrate the effect of multiple TTM rounds on the alteration of CD4+ T and T subset phenotypes and their functions. The cumulative effect of multiple TTM treatments could promote immunity by attenuation of the senescent CD4+ T cells including CD4+28^null^, CD4+NKG2D+, CD4+28+NKG2D+ and specifically, CD4+28^null^NKG2D+. Moreover, TTM could reduce aberrant function of senescent CD4+ T cells via decreasing the pro-inflammatory cytokine, IL-17, produced by CD4+28^null^NKG2D+, especially in high NKG2D expression populations. Therefore, multiple rounds of TTM may be suitable as a complementary therapy for health maintenance and improvement of the immune system in elderly populations.

## Figures and Tables

**Figure 1 ijerph-18-03210-f001:**
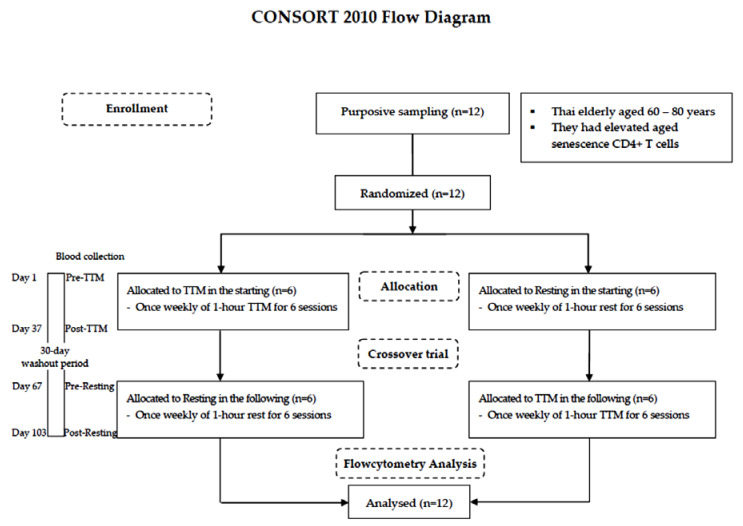
CONSORT 2010 flowchart diagram.

**Figure 2 ijerph-18-03210-f002:**
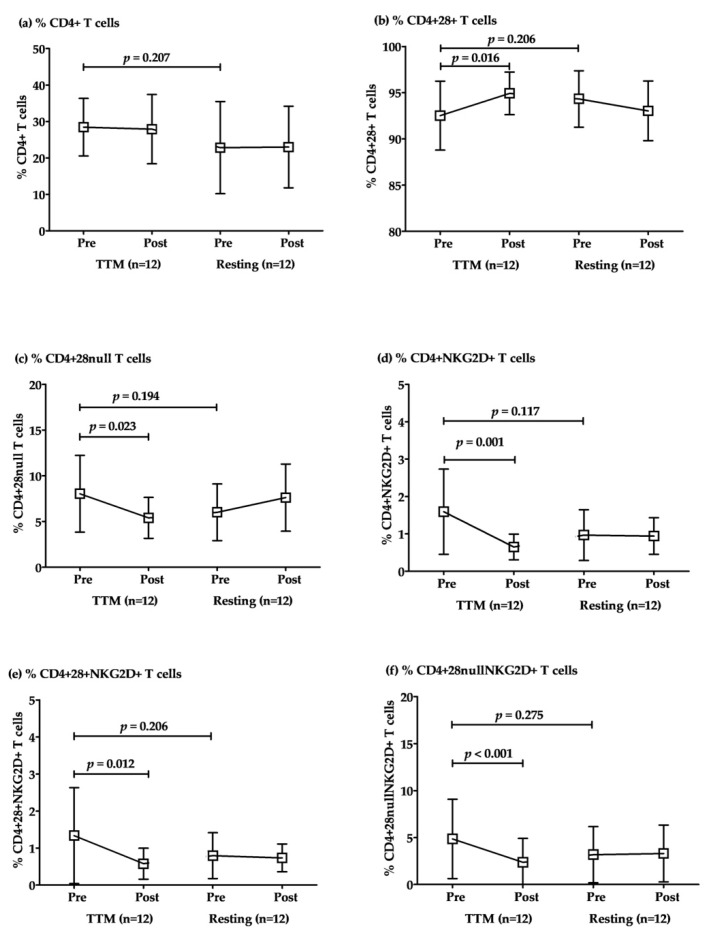
The alteration of CD4+ T cell subsets after multiple rounds of TTM. The representative plotted line graph displays the comparison of the percentages of CD4+ T (**a**), CD4+CD28+ T (**b**), CD4+CD28^null^ T (**c**), CD4+NKG2D+ T (**d**), CD4+CD28+NKG2D+ T (**e**), and CD4+CD28^null^NKG2D+ T cells (**f**) after TTM and resting. Data are displayed as mean ± SD. Statistical analysis was conducted by a two-way repeated-measures ANOVA with Bonferroni’s post hoc test, and significant differences were determined at *p* < 0.05.

**Figure 3 ijerph-18-03210-f003:**
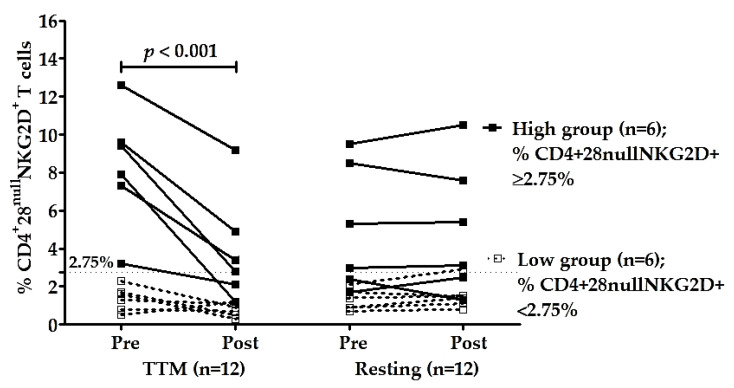
Changes in CD4+28^null^ NKG2D+ T cells of high group (*n* = 6) and low group (*n* = 6). Participants were divided based on the median of the percentages of CD4+28^null^ NKG2D+ T cells of pre-intervention (2.75%). The percentages of CD4+28^null^ NKG2D+ T cells of TTM and resting phase were analyzed between and within groups. Only the high group of the TTM phase showed a significant decrease in NKG2D expression on CD4+28^null^ T cells after multiple rounds of TTM (*n* = 6; *p* < 0.001), whereas others displayed no significant differences. Statistical analysis was through a two-way repeated-measures ANOVA with Bonferroni’s post hoc test. *p*-value < 0.05 was determined as a significant difference.

**Figure 4 ijerph-18-03210-f004:**
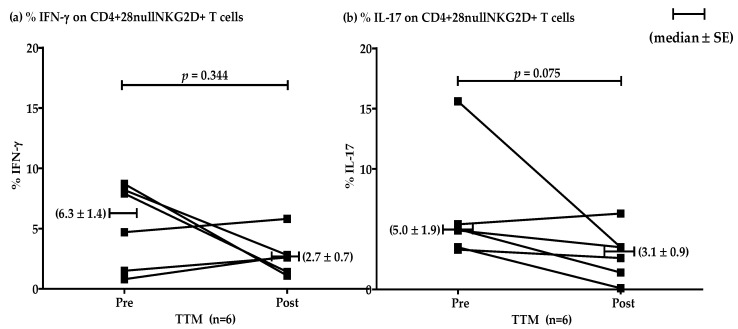
Representative plotting line graph (*n* = 6) displaying the comparison of the percentages of CD4+CD28^null^NKG2D+ T cells producing IFN-γ (**a**) and IL-17 (**b**) at pre- and post-TTM phase. The data are shown by median ± SE. IFN-γ production pre and post was 6.3 ± 1.4 and 2.7 ± 0.7, respectively. IL-17 production pre and post was 5.0 ± 1.9 and 3.1 ± 0.9, respectively. Data were analyzed by Wilcoxon Signed Rank test. *p*-value < 0.05 was determined as a significant difference.

**Table 1 ijerph-18-03210-t001:** Baseline characteristics of participants.

Parameters	Baseline (Mean ± SD, Range)	Reference Value
Number of samples (*n*)	12	
Gender (female/male)	9/3	
Age	67.7 ± 5.12 (61–75)	
Blood pressure with range		
Systolic (mmHg)	131.1 ± 10.8 (108–143) *	80–120
Diastolic (mmHg)	73.9 ± 10.2 (57–94)	60–80
Pulse rate	72.3 ± 7.7 (61–87)	80–100
HbA1c (NGSP), %	7.2 ± 2.4 *	4.6–6.2
Creatinine (mg/dL)	0.9 ± 0.2	0.5–1.5
ALT (U/L)	25.3 ± 13.0	4–36
Lipid profiles		
- TC (mg/dL)	217.3 ± 25.9 *	<200
- TG (mg/dL)	198.8 ± 175.0 *	<150
- HDL-C (mg/dL)	39.0 ± 15.2 *	>40
- LDL-C (mg/dL)	138.9 ± 33.5 *	<100
Senescent CD4+ T cell subsets (%) at baseline		
- CD4+28^null^	7.4 ± 4.6 (1.8–17.7)	
- CD4+NKG2D+	1.4 ± 1.2 (0.3–4.4)	
- CD4+28+NKG2D+	1.3 ± 1.4 (0.4–5.0)	
- CD4+28^null^NKG2D+	4.4 ± 4.3 (0.5–12.6)	

* abnormal values; SD: standard deviation; HbA1c: the glycated hemoglobin A1c; ALT: alkaline phosphatase; TC: total cholesterol; TG: triglyceride; LDL-C: low-density lipoprotein cholesterol; HDL-C: high-density lipoprotein cholesterol; NGSP: The National Glycohemoglobin Standardization Program. Notes: subjects had hypertension (2/12, 17%), DM (5/12, 42%) and dyslipidemia (3/12, 25%).

**Table 2 ijerph-18-03210-t002:** Comparisons of the percentages of CD4+ T cell subsets between groups at pre (baseline) and post (multiple sessions; once weekly, 6 sessions), and intergroups.

Parameters	Resting Group (*n* = 12)		TTM Group (*n* = 12)		Group × Time ^#^
	Pre	Post	Pre	Post	*p*
CD4+ T cell subsets (%)					
CD4+	22.8 ± 12.7	23.0 ± 11.2	28.4 ± 7.9	27.9 ± 9.5	0.960
CD4+28+	94.3 ± 3.1	93.0 ± 3.2	92.5 ± 3.7	94.9 ± 2.3 ^†^	0.035
CD4+28^null^	6.0 ± 3.1	7.6 ± 3.7	8.0 ± 4.2	5.4 ± 2.2 ^†^	0.039
CD4+NKG2D+	1.0 ± 0.7	0.9 ± 0.5	1.6 ± 1.1	0.7 ± 0.3 ^†^	0.044
CD4+28+NKG2D+	0.8 ± 0.6	0.7 ± 0.4	1.3 ± 1.3	0.6 ± 0.4 ^†^	0.084
CD4+28^null^NKG2D+	3.9 ± 3.0	3.3 ± 3.0	4.9 ± 4.2	2.4 ± 2.6 ^†^	0.006
High group of CD4+28^null^NKG2D+ (*n* = 6)	5.1 ± 3.3	5.1 ± 3.5	8.3 ± 3.1 *	4.0 ± 2.9 ^†^	<0.001

All data are expressed as means ± SD; ^#^ = analysis of two-way repeated-measures ANOVA; * = significant difference between groups at *p* < 0.05; ^†^ = significant difference between pre- and post-intervention at *p* < 0.05.

**Table 3 ijerph-18-03210-t003:** Effect sizes and post hoc power calculation.

Variable	Δ Change between Pre- and Post-TTM	*p*	Effect Size	Power (1-β Err Prob)
Effect of multiple round TTM				
CD4+	0.5 ± 1.6	0.866	0.33	0.17
CD4+28+	2.4 ± 1.4	0.016 ^†^	1.69	0.99
CD4+28^null^	2.6 ± 2.0	0.023 ^†^	1.35	0.98
CD4+NKG2D+	0.9 ± 0.8	0.001 ^†^	1.18	0.96
CD4+28+NKG2D+	0.8 ± 0.9	0.012 ^†^	0.86	0.77
CD4+28^null^NKG2D+	2.5 ± 1.7	<0.001 ^†^	1.49	0.99
High group of CD4+28^null^NKG2D+ (*n* = 6)	4.4 ± 0.3	<0.001 ^†^	17.6	1.00

The **Δ** change value was expressed as means ± SDs; difference between two dependent means (matched pairs) was analyzed by two-way repeated-measures ANOVA; ^†^ = Significant difference between pre- and post-intervention at *p* < 0.05; post hoc computed achieved power of type II error probability (1 − β) set at α = 0.05, two-tailed; each effect size and sample size (*n* = 12) using G*power 3.1 software.

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
