# Peer review of "Traditional Thai Massage Promoted Immunity in the Elderly via Attenuation of Senescent CD4+ T Cell Subsets: A Randomized Crossover Study"

_ijerph, 2021, doi:10.3390/ijerph18063210_

Round 1

Reviewer 1 Report

Interventions : 

Need to consider the order of treatment or not ? For example, did you need a group begin the upper limbs or a group begin the lower limbs ?

Was therapist the same person or not ? If there are multiple therapists, will the effect of Thai massage vary ?

Participant characteristics :

Is it not necessary to have the same number of male and female ? Is there a gender differences inn the effect of Thai massage ?

Are the subjects taking medication ? It seems that the subject has diabetes et al. Is it the effect of pure Thai massage ?

Author Response

We are most grateful to the reviewers' comments and suggestions. Accordingly, the manuscript has been improved tremendously. The responses and rectifications of the manuscript according to the comments and suggestions are attached.

Reviewer 2 Report

The manuscript by Sornkayasit et al. “Traditional Thai Massage Promoted Immunity in the Elderly via Attenuation of Aged Senescence CD4+ T Cell Subsets: A Randomized Crossover Study“ describes a positive effect of TTM treatment on the peripheral biomarkers of immunosenescence in elderly individuals. Using randomized controlled crossover design with a 30 days washout period, the authors demonstrated a significant decrease in late-differentiated CD4+ T-cell subsets especially those expressing activating molecules on their surface after the TTM-treatment, as well as some tendency in decrease in intracellular production of the inflammatory cytokines. The topic of the investigation is very interesting, but there is some room for improvement of the manuscript.

In general, the language of the manuscript is not always clear, making it difficult to follow – it is to be revised by native speaker.

Abstract:

Line 23-24: “Flow cytometric analysis of surface and cytokine intracellular staining was performed”. – It was presumably meant: … analysis of surface antigens expression? Please clarify and reformulate.

Lines 20, 21, 24: “aged senescence CD4+ T cell subsets” should be rather “senescent CD4+ T-cell subsets”?

Lines 25-26: “The participants were analyzed into 2 groups” – is apparently meant that participants were divided/allocated into 2 groups?

Lines 27-28: “…the high group had dramatically decreased in the cell population…” is presumably meant, that rather the cell population had dramatically decreased?

Introduction:

Lines 40-48: This paragraph, in which the authors intended to describe the previous studies, is unfortunately incomprehensible and consists mostly of incomplete interrupted sentence fragments, which do not provide the reader with any concrete information about the previous studies.

Line 53: “The aged-associated decline” – please change to “The age-associated decline”

Material and Methods

The methods are not described clearly and detailed enough for other researchers to replicate. The technical details, should be expanded and clarified to ensure that readers understand exactly, how experiments were done. For example, the immune cell analysis by flow cytometry is not described sufficiently: it is not clear, which markers were used for staining, and how the gating strategy for both cell surface and intracellular staining was applied. The figures for the gating strategy could be included in the Supplementary material section.

Line 84 and throughout the whole text: please correct the “aged senescence CD4+ T cells” into either “aged” or “senescent”.

Lines 114-115: “Effectiveness of acupuncture on lymphocytes” – should be rather “Effectiveness of acupuncture on lymphocyte proliferation”?

Lines 143-144: “Functional assay for stimulating T cells of 7 mL blood was activated by 100 ng/mL of phorbol-12-myristate- 13-acetate (PMA) and 1000 ng/mL ionomycin calcium salt (Sigma Aldrich, USA).” - Not intelligible sentence – please rewrite.

Lines 158-159: “Changes in intracellular cytokines…” should be rather “Changes in concentrations of intracellular cytokines…”?

Statistics:

The description of the statistical tests is very sparse. It is not entirely clear whether the authors used a mixed-models ANOVA or the traditional repeated-measures ANOVA. If it was the mixed-models ANOVA as they describe, then more information should be reported, e.g., what about random and fixed effect terms, how were they defined and were they significant or not.  What about the model fit statistics? And most importantly, regardless of what type of ANOVA was used, the factors and factor levels should be explicitly stated.

Results:

The description of the study design also mentioned assessment of other blood parameters, such as cholesterol and triglycerides. It is known that these blood markers and especially LDL play an important role in age-related processes, such as low-grade inflammation and inflammaging, respectively. What was the impact of the TTM on these blood parameters? Have they improved either? If not – please discuss this result.

The blood pressure was also measured – which effects of TTM on the blood pressure were found? If no effects were found – could you discuss this please.

Lines 210-211: “…the percentages of IFN-γ and IL-17 productions…” should be presumably “…the percentages of IFN-γ and IL-17 producing T cells…”?

Lines 216-217: The title of Table 2 is not intelligible – please correct

Lines 243-244: “the comparison of the percentages of IFN-γ (a) and IL-17 (b) produced by CD4+CD28nullNKG2D+ T cells” was apparently rather meant: “the comparison of the percentages of CD4+CD28nullNKG2D+ T cells producing IFN-γ (a) and IL-17 (b)”? The same problem in the headings to Fig. 4

Discussion:

The discussion should be rewritten. Some fragments of the discussion are hard to understand. Some parts of the discussion are not very relevant and can be omitted, while the other important results are not discussed at all. Here just an example, where results obtained in the present study are not sufficiently discussed.  

  • Why the positive effects of TTM are more pronounced in the group of elderly individuals with high percentage of the senescent CD4+ cells expressing stimulating molecules?
  • Which mechanisms might be responsible for this effect?

The authors conclude that the TTM is capable to improve the immune system. This statement is too broad and too imprecise and needs some more concrete ideas about what mechanisms and what actors may play a role in the background.

In general, I would not use the term “pathogenic CD4+ T cells”- even for the functionally senescent T cells.

This sentence is also difficult to understand: “Classification of CD4 T cell subsets in immunosenescence (the reduction of immunity in the elderly from advancing age) and immune disorders has documented the loss of CD28 expression (CD28- or CD28null) 267 [14], [16], [18], [22–24] and/or the augmentation of NKG2D expression (NKG2D+) [19], 268 [20], [25].” I would propose to redraft it.

Author Response

(The authors gave the same response as above.)

Round 2

Reviewer 2 Report

In my opinion, it is highly recommended to have the text of the manuscript corrected by a competent professional translator in English.

Author Response

Response to reviewer #2

  1. In my opinion, it is highly recommended to have the text of the manuscript corrected by a competent professional translator in English.

Response:       We would like to thank the reviewer for the comments and suggestions. We have submitted our manuscript for English editing process via MDPI as attached.